


# Modelling the climate and surface mass balance of polar ice sheets using RACMO2, part 2: Antarctica (1979–2016).

J. Melchior van Wessem[1], Willem Jan van de Berg[1], Brice P. Y. Noël[1], Erik van Meijgaard[2], Gerit Birnbaum[3], Constantijn L. Jakobs[1], Konstantin Krüger[3], Jan T. M. Lenaerts[4], Stef Lhermitte[5], Stefan R. M. Ligtenberg[1], Brooke Medley[6], Carleen H. Reijmer[1], Kristof van Tricht[7], Luke D. Trusel[8], Lambertus H. van Ulft[2], Bert Wouters[1], Jan Wuite[9], and Michiel R. van den Broeke[1]

[1]Institute for Marine and Atmospheric Research Utrecht, Utrecht University, Utrecht, the Netherlands
[2]Royal Netherlands Meteorological Institute, De Bilt, the Netherlands
[3]Alfred Wegener Institute, Helmholtz Centre for Polar and Marine Research, Bremerhaven, Germany
[4]Department of Atmospheric and Oceanic Sciences, University of Colorado, Boulder CO, USA
[5]Department of Geoscience and Remote Sensing, Delft University of Technology, Delft, The Netherlands
[6]Cryospheric Sciences Laboratory, NASA Goddard Space Flight Center, Greenbelt, MD, USA
[7]KU Leuven, Department of Earth and Environmental Sciences, Leuven, Belgium
[8]Department of Geology, Rowan University, Glassboro, New Jersey, USA
[9]ENVEO IT GmbH, Innsbruck, Austria

*Correspondence to:* J. M. van Wessem (j.m.vanwessem@uu.nl)

**Abstract.** We evaluate modelled Antarctic ice sheet (AIS) near-surface climate, surface mass balance (SMB) and surface energy balance (SEB) from the updated polar version of the regional atmospheric climate model RACMO2 (1979–2016). The updated model, referred to as RACMO2.3p2, incorporates upper-air relaxation, a revised topography, tuned parameters in the cloud scheme to generate more precipitation towards the AIS interior, and modified snow properties reducing drifting snow

sublimation and increasing surface snowmelt.

Comparisons of RACMO2 model output with several independent observational data show that the existing biases in AIS temperature, radiative fluxes and SMB components are further reduced with respect to the previous model version. The model integrated annual average SMB for the ice sheet including ice shelves (minus the Antarctic Peninsula (AP)) now amounts to 2229 Gt y$^{-1}$, with an interannual variability of 109 Gt y$^{-1}$. The largest improvement is found in modelled surface snowmelt,

that now compares well with satellite and weather station observations. For the high-resolution (∼5.5 km) AP simulation, results remain comparable to earlier studies.

The updated model provides a new, high-resolution dataset of the contemporary near-surface climate and SMB of the AIS; this model version will be used for future climate scenario projections in a forthcoming study.

## 1 Introduction

Before being able to accurately predict future changes in the climate and surface mass balance (SMB) of the Antarctic ice sheet (AIS), it is required that its contemporary climate and SMB are realistically modelled. The interaction of the ice sheet with its atmospheric environment can be studied with regional climate models (RCMs) that are specifically adapted for use over



the polar ice sheets of Greenland (Noël et al., 2016; Langen et al., 2017; Fettweis et al., 2017) and Antarctica (Gallée et al., 2013; Van Wessem et al., 2014b). Such polar RCMs explicitly calculate the individual components of the ice sheet SMB, such as precipitation, snow sublimation and surface meltwater runoff into the ocean. Combining integrated SMB with estimates of glacial discharge (D) then gives the mass balance of the ice sheet (MB) and its associated contribution to global sea level

change (Rignot et al., 2011; Shepherd et al., 2012).

Over Antarctica, precipitation is the dominating component of the SMB, contributing 91% to the total (the sum of the absolute fluxes) mass budget (Van Wessem et al., 2014b). Snowfall rates are expected to rise in the future as a result of the increased moisture holding capacity of a warming atmosphere (Ligtenberg et al., 2013), which potentially compensates (Barrand et al., 2013) or amplifies (Winkelmann et al., 2012) changes in glacial discharge rates.

Surface melt rates along the coastal margins of the ice sheet can be significant, further amplified by local climate features (Lenaerts et al., 2016b), possibly leading to ice shelf hydrofracturing (Van den Broeke et al., 2005a; Scambos et al., 2009; Kuipers Munneke et al., 2014), and the acceleration of grounded glaciers and related sea level rise (Rignot, 2004).

Polar RCMs are required to model above processes realistically, and, in this study, the Regional Atmospheric Climate MOdel (RACMO2) is used at 27 km spatial resolution to simulate the climate and SMB of Antarctica. RACMO2 is able to accurately

simulate climate variables over Antarctica (Van Lipzig et al., 2002; Reijmer et al., 2005; Van de Berg et al., 2006; Lenaerts et al., 2012b). However, substantial challenges remain. For instance, RACMO2 systematically underestimates snowfall, and hence the SMB, over the East Antarctic plateau. This negative bias was first identified in Van de Berg et al. (2006) but persisted in subsequent model versions at ∼10% (Van Wessem et al., 2014b). In addition, biases in the net longwave radiation and sensible heat fluxes (Van Wessem et al., 2014a), resulted in biases in the energy available for melt (King et al., 2015); or the

so-called surface energy budget (SEB), the sum of all energy fluxes at the surface. As a result, the previous model versions were systematically too cold over the ice sheet (Van den Broeke, 2008). In regions of melt, too little meltwater percolates into and refreezes in the snow column, resulting in too low snow temperatures (Van den Broeke, 2008). Although this bias was substantially reduced, it still remains at –1.3 K in the latest model version (Van Wessem et al., 2014a).

One of the main causes of these discrepancies are shortcomings in the cloud microphysics. Biases in SMB and SEB are

caused by too thin clouds simulated over the AIS, resulting in too little snowfall, too much downwelling shortwave radiation and too little downwelling longwave radiation (Van Wessem et al., 2014a; King et al., 2015). In addition, these biases are potentially related to an unrealistic fractionation of ice and water content in these clouds, which significantly affects the sensitivity of the (cloud) radiative fluxes to changes in cloud content (King et al., 2015).

Another model issue is the relatively coarse horizontal resolution of RACMO2. A high spatial resolution is important to

resolve the interaction of the topography with the atmosphere in detail, realistically simulating topography related processes such as katabatic winds and orographically forced precipitation (Genthon and Krinner, 2001). Since Van de Berg et al. (2006) and Van Lipzig et al. (2002) the horizontal resolution has been refined from 55 km to 27 km in Lenaerts et al. (2012b), and recently climate simulations at 5.5 km resolution have been performed, focusing on specific regions such as the Antarctic Peninsula (AP) (Van Wessem et al., 2016), Dronning Maud Land (Lenaerts et al., 2016a), the West-Antarctic coast (Lenaerts

et al., 2017, in preparation) and Adèlie Land (Lenaerts et al., 2012a). Limitations in the model topography also result from



biases in the source datasets used for aggregation (Liu et al., 2001; Bamber and Gomez-Dans, 2009). These datasets are typically based on observational data from remote sensing techniques such as airborne (DiMarzio et al., 2007) and satellite measurements (Rignot et al., 2008), but frequent improvements are applied to these datasets as well (Griggs and Bamber, 2009; Borsa et al., 2014).

Here, we discuss the effects of an update from RACMO2.3 version p1 to version p2, which addresses the model challenges presented above. To assess whether this update improves the modelled climate of the AIS, in terms of SEB and SMB, we revisit several of the evaluations done in previous studies. First, Section 2 discusses the model, the updates included and the observational datasets used for model evaluation. Section 3 presents the changes in terms of simulated SEB and SMB for the full ice sheet, as well as the evaluation by comparison with observations. Section 4 presents the specific changes and evaluation

for the 5.5 km model version applied to the AP. Section 5 then discusses remaining challenges and model limitations, followed by a summary and conclusions in Section 6. This manuscript is part of a tandem model evaluation over the Greenland (part 1) and Antarctic (this study) ice sheets.

## 2   Model and observational data

### 2.1   The Regional Atmospheric Climate Model RACMO2

In this study, we use the Regional Atmospheric Climate MOdel version 2.3 (RACMO2.3). The model combines the atmospheric dynamics of the High Resolution Limited Area Model (HIRLAM, Undén et al., 2002) and the physics package CY33r1 of the European Centre for Medium-Range Weather Forecast (ECMWF) Integrated Forecast System (IFS) (ECMWF-IFS, 2008). The model assumes hydrostatic equilibrium and we have verified that at both horizontal resolutions (27 and 5.5 km) the hydrostatic assumption provides realistic results (Van Wessem et al., 2015, 2016).

This version of the model is specifically applied to the polar regions by interactively coupling it to a multilayer snow model that calculates melt, refreezing, percolation and runoff of meltwater (Greuell and Konzelmann, 1994; Ettema et al., 2010). In addition, snow albedo is calculated through a prognostic scheme for snow grain size (Kuipers Munneke et al., 2011) while a drifting snow scheme simulates the interaction of the near-surface air with drifting snow (Déry and Yau, 1999; Lenaerts et al., 2010). Throughout this study, two model domains are addressed: RACMO2.3/ANT simulates the climate of the full Antarctic

ice sheet (AIS), while RACMO2.3/AP is applied to the AP region specifically (Figure 1).

### 2.2   Surface energy budget and surface mass balance

RACMO2.3 explicitly resolves surface melting by solving the surface energy budget (SEB; W m$^{-2}$), defined as:

$$M = SW_d + SW_u + LW_d + LW_u + SHF + LHF + G_s, \qquad (1)$$

where fluxes directed towards the surface are defined positive, M is melt energy, SW and LW are the (upward- and downward)

shortwave and longwave radiative fluxes, SHF and LHF are the sensible and latent turbulent heat fluxes and $G_s$ is the subsurface





conductive heat flux. Excess energy at the surface is used to produce meltwater which can percolate through the snow column, where it is refrozen or retained. Ultimately, when the snowpack is saturated, the meltwater runs off to the ocean, representing a negative component in the surface mass balance (SMB) of the ice sheet. The SMB, in $\mathrm{mm}$ w.e. $\mathrm{y}^{-1}$, is defined as:

$$\mathrm{SMB} = \mathrm{P}_{\mathrm{tot}} - \mathrm{SU}_{\mathrm{s}} - \mathrm{SU}_{\mathrm{ds}} - \mathrm{ER}_{\mathrm{ds}} - \mathrm{RU} \qquad (2)$$

where $\mathrm{P}_{\mathrm{tot}}$ represents total precipitation (snowfall (SN) plus rain (RA)), SU surface ($\mathrm{SU}_{\mathrm{s}}$) plus drifting snow ($\mathrm{SU}_{\mathrm{ds}}$) subli­mation, $\mathrm{ER}_{\mathrm{ds}}$ drifting snow erosion and RU meltwater runoff, the amount of liquid water (melt and rain) that is not retained or refrozen (RF) in the snowpack. Note that by defining SMB in this way, we include processes in the snowpack such as refreezing; this is formally referred to as the climatic SMB (Cogley et al., 2011).

## 2.3    Model updates

The model update includes small bugfixes and tuning of atmosphere and snow parameterisations, as summarized below. Throughout this paper we will refer to the new RACMO2.3 version as RACMO2.3p2, and to the old version as RACMO2.3p1, where *p* stands for *polar*.

a) Previous studies found that the previous model version systematically underestimates snowfall and downwelling long­
wave radiation in the interior of the ice sheets of both Greenland (Noël et al., 2015) and Antarctica (Van Wessem et al., 2014b). Therefore, the critical cloud water and cloud ice content ($\mathrm{l}_{\mathrm{crit}}$) thresholds governing the onset of effective precipitation forma­tion for mixed-phase and ice clouds, are increased by a factor 2 (Eqs. 5.35 and 6.39 in ECMWF-IFS (2008)) and 5 (Eq. 6.42 in ECMWF-IFS (2008)), respectively. As a result, we expect increased cloud cover for colder conditions and higher elevations, and precipitation simulated further inland. Consequently, we expect to further decrease SEB and SMB biases in the AIS inte­
rior, in a way similar to Van Wessem et al. (2014a, b).

b) The linear saltation snow load parameter (Equation 24, Déry and Yau, 1999) used in the drifting snow sublimation scheme is halved, i.e. from 0.385 to 0.190 (Lenaerts et al., 2012b), effectively halving the horizontal drifting snow mass and sublima­tion fluxes, without changing the length or frequency of drifting snow events, which were well simulated (Lenaerts et al.,
2012b). This makes the simulated drifting snow fluxes more in line with the limited drifting snow observations over Greenland (Lenaerts et al., 2014), and we include it for the Antarctic simulations as well.

c) The albedo of superimposed ice layers now follows Kuipers Munneke et al. (2011). Previously, the albedo used for su­perimposed ice was prescribed at 0.55, underestimating surface albedo, and overestimating melt. Although the effect of this
update is likely minor for Antarctica, where melting and the formation of superimposed ice layers in the current climate are





rather limited in occurence, this can result in better modelled surface albedo and melt locally, and in future simulations.

d) A number of minor model bugs, mostly related to the snow model and the resulting snowmelt fluxes, were fixed. Improvements were made in the snow grain evolution scheme, leading to faster snow grain metamorphism in the uppermost snow layer. As a result, albedo decay is enhanced, which in turn enhances snow melt. Two changes applied to the RACMO2 version used over Greenland have not been included in RACMO2/ANT: 1) the size of refrozen snow grains is reduced from 2 mm to 1 mm (this value was already used in RACMO2/ANT), and 2) the model soot concentration, consisting of dust and black carbon impurities deposited on snow, has been halved from 0.1 ppmv to 0.05 ppmv. Over Antarctica, likely no soot is present and model soot content is kept at zero.

## 2.4 Initialisation and set up

RACMO2.3 uses a vertical mesh of 40 hybrid model layers, and a horizontal resolution of 27 km over the AIS and of 5.5 km over the AP. Both model versions are forced at the lateral boundaries (Fig. 1) by ERA-Interim reanalyis data every 6 hours (Jan. 1979 – Dec. 2016, Dee et al., 2011). RACMO2.3 is not coupled to an ocean model and sea surface temperature and sea ice cover are prescribed from the reanalysis. The lateral boundaries are now taken from higher resolution ($0.75°$) ERA-interim fields than before ($1.5°$), mostly affecting sea-ice near the ice sheet margin.

RACMO2.3 is coupled to a snow model, a single column time-dependent model that describes the evolution of the firn layer, based on a firn densification model (IMAU-FDM). It calculates firn density, temperature and liquid water content evolution based on forcing at the surface by surface temperature, accumulation and wind speed. Surface meltwater percolates into the model firn layer, where it can refreeze, be stored or percolate further down. The retention of meltwater is based on the tipping-bucket method (i.e. liquid water is stored in the first available layer and transported downwards only when it exceeds the maximum capillary retention). Liquid water that reaches the bottom of the firn layer is removed as runoff. More details on the snow model can be found in Ettema et al. (2010), Ligtenberg et al. (2011) and Kuipers Munneke et al. (2015).

Both RACMO2.3/ANT and RACMO2.3/AP are initialised at 1 January 1979; the initial firnpack is taken from a simulation with an offline IMAU-FDM (Ligtenberg et al., 2011), which has the same physics but a higher vertical resolution than the internal snow model, and is driven by the previous RACMO2.3 climatologies for both model versions (Van Wessem et al., 2014b, 2016).

### 2.4.1 Specific RACMO2.3p2/ANT set up changes

a) The model topography in earlier model versions was aggregated from the digital elevation model (DEM) from Liu et al. (2001). The updated topography is based on the (1 km spatial resolution) DEM from Bamber and Gomez-Dans (2009), derived from a combination of satellite radar and laser data. As a result, the topography in the coastal regions is now better resolved (Griggs and Bamber, 2009), likely resulting in better modelled topography related variables, such as melt, katabatic winds and orographic precipitation.



b) We apply boundary relaxation by ERA-Interim wind fields in the upper atmosphere, following Van de Berg and Medley (2016). As a result, interannual variability is improved in the domain interior. Annual average changes are small (Van de Berg and Medley, 2016).

### 2.4.2 Specific RACMO2.3p2/AP set up changes

a) The DEM used in RACMO2.3/ANT was already used for the region south of 70°S (Van Wessem et al., 2016). This topography remains unchanged in the new model version. However, a projection parameter incorrectly used in the previous simulations has been changed, slightly altering the aggregation of the DEM to the model topography, resulting in a northward (∼5 km; one grid box) displacement of topographic features.

b) The model integration domain is extended by 60 grid points to the north of the original RACMO2.3/AP domain, allowing the inclusion of the South Shetland Islands and Elephant Island well into the domain (see Fig. 1).

c) Upper-air relaxation in RACMO2.3/AP is not used, as the model domain is smaller and variability therefore better constrained at the lateral boundaries, and because the wind relaxation would smooth out orographic precipitation over the detailed AP topography.

## 2.5 Observational data

We evaluate the updated model by revisiting the comparisons of modelled SMB components and SEB done in previous studies (Van Wessem et al., 2014a, b, 2015, 2016). These observational data include SEB data from Automatic Weather Stations (AWS), 10 m snow temperature, in-situ SMB observations, accumulation radar profiles, and solid ice discharge estimates. Detailed descriptions of these methods are provided in their respective papers. Here, we describe updates to the observational datasets only.

### 2.5.1 Automatic Weather Stations

We evaluate modelled near-surface wind, temperature and SEB components using nine AWSs located in Dronning Maud Land (DML) in different climate regimes. These AWSs measure all four radiation components as well as humidity and snow temperature, which are used to force an Energy Balance Model (EBM), effectively closing the SEB at AWS locations. A summary of the location and data records of the AWSs is provided in Table 1 and locations are shown in Fig. 1. We use the same time period (1979–2011) for evaluation as in Van Wessem et al. (2014a) and did not extend the few AWSs that have longer records. For more details on the EBM see Van den Broeke et al. (2005a, b) and Reijmer and Oerlemans (2002). All data from AWSs are monthly averaged and compared with data from the same months from RACMO2.3.



In addition to these AWSs, in order to evaluate the modelled melt fluxes, we have included SEB data from Neumayer station in DML that span a longer time period (1990–2015) (Van den Broeke et al., 2009). We use these data to force the EBM, and test the sensitivity of the calculated melt to two different surface roughness lengths: $z_0 = 28$ (a commonly used setting) mm and $z_0 = 1$ mm (standard setting in the EBM).

### 2.5.2 In-situ SMB and temperature observations

We compare the updated RACMO2.3/ANT modelled SMB with 3234 in-situ SMB observations (Fig. 1) described by Favier et al. (2013). For modelled surface temperature evaluation, we use the 64 snow temperature observations (Fig. 1) used in Van Wessem et al. (2014a).

### 2.5.3 Accumulation radar

In order to evaluate model performance in capturing the temporal variability in SMB, we use radar-derived annual accumulation rates from 1980 to 2009 over much of the Thwaites glacier catchment area generated by Medley et al. (2013). The accumulation rates were derived from the Center for Remote Sensing of Ice Sheet snow radar system as part of NASA Operation IceBridge (Leuschen, 2014). Similarly, we calculate accumulation rates using snow radar data from two additional surveys (Figure 1): the western Getz ice shelf (17 October 2011) and the Ronne ice shelf (15 November 2015) near the Institute Ice Stream grounding zone. The areas covered by these new surveys are much smaller than the Thwaites glacier survey (∼1600 km), but are larger than the RACMO2.3 grid cell width (Getz: ∼75 km, Ronne: ∼50 km), averaging out much of the spatial variability. We use a method described in Medley et al. (2015), which allows for spatial variations in the firn density profile in both radar-depth estimation as well as conversion to water equivalence. The method iteratively solves for a depth-density profile for each measurement that is consistent with the radar-derived accumulation rate as well as long-term modeled 2-meter air temperature from MERRA-2 and an initial density of $350 \, \mathrm{kg \, m^{-3}}$. For the Ronne ice shelf survey, we are able to generate a 30-year time series (1985–2014); however, we are only able to record 10 years of accumulation over the Getz ice shelf, which is due to its high accumulation rate ($\sim 900 \, \mathrm{mm \, w.e. \, y^{-1}}$). Thus, the model evaluation over the Getz is less robust than over the Ronne and Thwaites surveys.

### 2.5.4 GRACE

The Gravity Recovery and Climate Experiment (GRACE) mission provides monthly observations of mass changes across the AIS at a resolution of ∼300 km. To assess temporal simulated SMB variability, we regionally compare cumulative SMB with the GRACE observations at a regional scale. The GRACE data (CSR RL05) are processed as in Van Wessem et al. (2014b). Mass anomalies are assigned to predefined drainage basins (Zwally et al., 2012). To better capture the higher SMB variability in the marginal zones, these basins are split up in coastal and interior sub-basins. The assigned anomalies are then converted to pseudo-GRACE observations and adjusted until the differences with the actual GRACE observations are minimized in a least-squares sense. GRACE measures the integral sum of mass anomalies along the satellite orbit. Atmospheric and oceanic





mass variability is removed by the processing centers using numerical models. However, residual signals may remain due model deficiencies, mostly at high frequencies. To reduce aliasing effects of these signals, both the GRACE and RACMO2.3 data are smoothed with a seven-month moving average filter. As GRACE measures the sum of SMB and ice dynamics, which generally act at slower time scales, all time series are detrended and a quadratic acceleration term is removed.

### 2.5.5 QuikSCAT meltfuxes

We use estimates of surface meltwater production from the satellite radar backscatter time series from QuikSCAT (QSCAT), which is calibrated with SEB-derived melt flux observations (Trusel et al., 2013). We use these to compare with modelled melt fluxes for the period 2000-2009 for the ice shelf regions denoted in Figure 1.

### 2.5.6 Cloudsat-CALIPSO

To evaluate the modelled downwelling radiative fluxes we use a modified version of Release 04 (R04) of the CloudSat Level 2B Fluxes and Heating Rates (2B-FLXHR-LIDAR) product (Henderson et al., 2013) with specific adaptations for the polar atmosphere (Van Tricht et al., 2016b), combining cloud properties retrieved by Cloudsat-CALIPSO (C-C), reanalysis data from ERA-Interim, and surface properties, which drives a broadband radiative transfer model. Data from 2007–2010 is available at a horizontal resolution of 2°by 1°as described in Lenaerts et al. (2017); Matus and L'Ecuyer (2017). The performance of this product over polar regions has been evaluated by a statistical comparison with AWS measurements both in the Arctic as well as in the Antarctic (Van Tricht et al., 2016a, b). Root mean square deviations (RMSD) of the computed fluxes and the AWS measurements from the four Antarctic AWSs are used as uncertainties on the fluxes used in this study.

### 2.5.7 Upper air profiles

To evaluate modelled (upper) atmospheric conditions we use radiosonde data from Kohnen station, close to AWS 9, maintained by the Alfred Wegener Institute (AWI). As part of austral summer campaigns at Kohnen Station (75°S, 0°E, 2892 m a.s.l.), in 2005/06 and 2013/14, radiosonde measurements were performed four times a day (00 UTC, 06 UTC, 12 UTC, 18 UTC). Radiosondes of type RS92-SGP manufactured by VAISALA, Finland, were used. The sondes carried sensors to measure pressure, temperature, humidity and a GPS receiver to capture wind speed and wind direction. At 06 and 18 UTC a 200 g balloon was launched. At 00 and 12 UTC a 600 g balloon was utilized, which reached altitudes up to 35 km above sea level. The radiosondes transmitted vertical data in time steps of 2 s. Hence, we obtain profile data with a vertical resolution of about approximately 10 m, which provide the basis for the evaluation of RACMO2.3/ANT data. Here, we use data from austral summer 2013/2014 which are presented in section 3.4.4; results for 2005/2006 are similar.

### 2.5.8 Discharge estimates

To evaluate RACMO2.3/AP modelled SMB we use the solid ice discharge estimates from Wuite et al. (2015), as also used in Van Wessem et al. (2016). We added similar discharge estimates for George IV ice shelf (Hogg et al., 2017).



## 3 Results: full ice sheet at 27 km

### 3.1 Changes in modelled SEB

Figure 2 shows the difference in modelled annual mean (1979–2014) SEB and near-surface variables between RACMO2.3p2 and RACMO2.3p1. Figure 2a shows a relatively uniform increase of $LW_d$ of up to 10 W m$^{-2}$, a result of the changes in the

5 model cloud scheme, leading to more clouds inland. The increase is largest over the sea-ice and along the coastal margins and the escarpment zone of the ice sheet, where cloud coverage peaks. Figure 2b illustrates that surface temperature $T_s$ has increased uniformly as a result of the increase in $LW_d$. At the East Antarctic plateau changes are smaller. Differences in $V_{10m}$ are small and most pronounced in regions of steep topography, where changes in the model surface topography are largest, but are also due to changes in the near-surface temperature inversion $T_{inv}=T_s-T_{2m}$ as a result of the changes in $T_s$. Finally, Figure

2d shows the resulting changes in the sensible heat flux (SHF), which are closely related to the changes in $LW_{net}$, causing SHF to drop as the surface temperature inversion is weaker. Figure 2 also shows sharp local changes over polynyas between the ice-shelf edge and the sea-ice. These differences are mostly related to changes in the ice-mask, resulting from the updated surface topography, but are also related to the higher-resolution ocean and sea-ice boundary forcing. The latter change has caused sea-ice to be better resolved along the coastal margins, with associated changes in the surface fluxes.

### 3.2 Changes in modelled SMB

Figure 3 shows the related changes in the SMB components. First, the patterns in the change in total precipitation (snowfall+rainfall, Fig. 3a) are comparable to the changes in $LW_d$ and $T_s$: there is a migration of precipitation from the ocean and coastal margin towards the interior of the ice sheet due to the updated cloud scheme. However, in a few coastal regions with pronounced topography there is a decrease related to the updated model topography. For instance, enhanced precipita-

20 tion shadow patterns are found in the Antarctic Peninsula (AP), parts of Dronning Maud Land (DML), and Marie Byrd Land (MBL). Figure 3b illustrates that sublimation fluxes are smaller locally, removing up to 200 mm w.e. y$^{-1}$ less mass. This is a result of the lowered snow drift saltation parameter, causing drifting snow sublimation $SU_{ds}$ to be less efficient in the coastal and escarpment region where wind speeds and $SU_{ds}$ are the largest. Over the flat ice shelves, as well as some regions where the new topography has resulted in lower elevations, warmer conditions have resulted in a small increase of (surface) sublimation.

Surface snowmelt fluxes (Fig. 3c) increased by $\sim$100 mm w.e. y$^{-1}$ over the coastal region and ice shelves. The changes are mainly caused by the faster snow grain growth, which lowers surface albedo and enhances summer snowmelt. Surface melt has no influence on the SMB, as nearly all meltwater refreezes in the firn. Ultimately, changes in SMB (Fig. 3d) are dominated by changes in $P_{tot}$, which are the largest.

### 3.3 Changes in integrated SMB

Table 2 summarizes the RACMO2.3 integrated values of the SMB components, calculated for the model ice mask including ice shelves, but excluding the AP. In RACMO2.3p2/ANT integrated SMB amounts to 2229 Gt y$^{-1}$ with an interannual





variability of $\sigma = 109$ Gt y$^{-1}$, an increase of 69 Gt y$^{-1}$ (3.2%). Changes in precipitation are mainly caused by a redistribution over the ice sheet; integrated changes over the total ice sheet are small: P$_{tot}$ has increased slightly by 14 Gt y$^{-1}$ (0.5%) to 2400 $\pm 109$ Gt y$^{-1}$. The increase in SMB is mostly caused by a reduction in drifting snow sublimation SU$_{ds}$, which has dropped by 79 Gt y$^{-1}$, from $181 \pm 9$ Gt y$^{-1}$ to $102 \pm 5$ Gt y$^{-1}$. An increase of 22 Gt y$^{-1}$ in surface sublimation partly

compensates this decrease, total SU has decreased by 56 Gt y$^{-1}$ to $161 \pm 7$ Gt y$^{-1}$. Snowmelt has increased significantly from $36 \pm 17$ Gt y$^{-1}$ to $71 \pm 28$ Gt y$^{-1}$, an increase of 97%. The increase does not affect the SMB, as all extra meltwater (and rain) is modelled to refreeze in the snow ($71 \pm 28$ Gt y$^{-1}$). The remaining SMB components (rainfall, ER$_{ds}$ and runoff) have not changed significantly.

Integrated SMB for the grounded ice sheet (GIS), which is based on the Rignot et al., 2017 (in preparation) drainage basin

definitions excluding the AP, has increased by 103 Gt y$^{-1}$ to 1885 Gt y$^{-1}$, which is a bigger increase than when including the ice shelves. This difference is caused by a redistribution of snowfall towards the interior of the ice sheet (see Fig. 3a,d) and because changes in SU$_{ds}$ are mainly present over grounded ice (Fig. 3b). This effect is most profound over the East Antarctic ice sheet (EAIS), where the SMB has increased by 79 Gt y$^{-1}$ (7.5%), compared to an increase of only 17 Gt y$^{-1}$ (2.7%) over the West Antarctic ice sheet (WAIS).

### 3.4 Evaluation of modelled SEB

#### 3.4.1 Automatic Weather Stations

Figure 4 compares monthly modelled SEB components with AWS observations. All components show consistent improvements. For net longwave radiation (LW$_{net}$), the correlation (r$^2$) has increased from 0.69 to 0.77, the bias decreased from –6.9 W m$^{-2}$ to –4.5 W m$^{-2}$ and the RMSD decreased from 7.8 W m$^{-2}$ to 7.0 W m$^{-2}$. Similar but less pronounced improvements

are found for the other components, most notably SHF. Improved correlation is mostly caused by the inclusion of upper air relaxation in the model: temporal variability is now better constrained (Van de Berg and Medley, 2016). Improvements in bias and RMSD are mainly caused by changes in the cloud scheme.

Figure 4e shows small improvements for V$_{10m}$. Temporal variability has improved in a similar way as for the SEB, but improvements in slope, bias and RMSD are mostly unrelated to the improvements in the SEB components. These changes are

related to the new model topography: the regression slope of V$_{10m}$ has gone up considerably from 0.43 to 0.56. This poor fit is generally caused by the relatively coarse horizontal resolution, resulting in an underestimation of surface slope in steep areas, and an overestimation in flat areas (Reijmer and Oerlemans, 2002; Van Wessem et al., 2014a). Therefore, katabatic winds that are strongly related to the surface slope are too weak/strong in sloped/flat regions. Even though the horizontal resolution in RACMO2.3/ANT is unchanged, surface slope is likely better represented in the new model topography. The improvement in

T$_s$ (see below) also partly leads to a better representation of the surface temperature inversion T$_{inv}$, which affects the strength of the katabatic wind forcing.

Figure 4f shows modelled T$_s$, as a function of observed T$_s$ calculated by closing the SEB. A general shift to higher temperatures is seen for the entire temperature range, but temperature is still underestimated in most months. This warming results in



overall better statistics, although at some locations temperature now is/remains overestimated. This overestimation overcompensates some of the biases, which is reflected in the change of the RMSD: while the absolute bias suggests a near perfect match of modelled and observed $T_s$ (–0.14K), RMSD shows a more modest improvement by 0.26 K.

### 3.4.2   10 m firn temperature

To evaluate changes in $T_s$ over a larger area, Figure 5 shows the comparison of RACMO2.3/ANT modelled 10 m firn temperature with 64 observations. Slope and spatial correlation have remained similar, but the bias and RMSD have decreased significantly, by 1 K and 0.35 K respectively. The relatively small decrease of the RMSD is caused by the persistent positive temperature bias for the relatively cold East Antarctic plateau. At these locations the updated model now overestimates temperature by up to 3 degrees, likely related to the slight overestimation of downwelling longwave radiation with the updated cloud scheme.

### 3.4.3   Cloudsat-CALIPSO

Figure 6a shows the $LW_d$ and $SW_d$ bias and the observations (right axis) for all model ice sheet grid points averaged in nine surface elevation bins. For $SW_d$ the bias has decreased by about $\sim$5 W m$^{-2}$ for all elevation bins. Generally, over the AIS $SW_d$ remains overestimated, but the new model version simulates these fluxes better. Improvements in $LW_d$ are also evident: here the old model version underestimated $LW_d$ for all elevation bins, but more so at lower elevations ($<$1500 m). In the updated model the bias for these lower elevation bins, mostly representing the ice shelves and the WAIS, is reduced. However, for the elevation bins $>$2000 m, that represent the East Antarctic plateau, $LW_d$ is now overestimated by $\sim$4 W m$^{-2}$. Overall, the downwelling radiative fluxes are likely better represented in the new model version, but errors in these fluxes are large. Only for intermediate elevations, the third to fifth bins, the changes appear larger than the uncertainty in the fluxes, which is based on a comparison with ground-based AWS observations (Van Tricht et al., 2016a).

Unfortunately, figure 6b also shows that a systematic overestimation of both cloud ice and water is present in the new model version. Apparently, there is an error that compensates the biases in the downwelling radiative fluxes: by simulating clouds that are too optically thick, biases in the fluxes, and also in $P_{tot}$, have decreased. The general biases in these cloud variables appear relatively large, especially for the coastal bins. However, the biggest fraction of grid points is located in the high elevation bins as shown in the bar chart in Fig. 6a, where biases are the lowest.

### 3.4.4   Kohnen station radiosondes

To assess upper atmosphere conditions, Figure 7 compares modelled temperature, relative humidity and wind speed profiles with radiosonde measurements conducted at Kohnen station (75°S, 0°E, 2892 m a.s.l.). Here, we only show the average daily values for January 2014; the comparisons for other years give similar results. Figure 7 shows a consistent improvement in modelled temperature, relative humidity and wind speed. RMSD has decreased significantly for all three variables and at all height levels in the atmosphere, a direct result of the upper-air relaxation included in RACMO2.3p2. The improvement is



largest for wind speed; for temperature the changes are more variable near the surface (0–1000 m). Interesting is the significant improvement in simulated wind speed and temperature at the tropopause inversion ($\sim$ 8000 m). Here, RMSD is largest which is related to the maximum in wind speed and temperature at this level, and the difficulty in simulating the accurate vertical location of the inversion.

## 3.5 Evaluation of modelled SMB

### 3.5.1 In-situ observations

Figure 8 shows the SMB bias (model–observation) for both model versions averaged in nine surface elevation bins. The uniform increase of cloud cover and $P_{tot}$, accompanied by the decrease in drifting snow sublimation, increased SMB in all elevation bins. Only in the lowest bin, which represents the ice shelves, SMB decreased because precipitation now falls at higher elevations. The increase in SMB leads to a relative SMB bias that is generally below 5% for all bins above 250 m elevation. For some bins, SMB is even slightly overestimated. This change is consistent with those in temperature (Figs. 4f and 5b): more clouds in the interior result in higher temperatures and snowfall rates. The fifth and sixth elevation bins (1750–2750 m) show the largest improvements: negative biases decreased from – 9% and –16% to 1% and –6%, respectively. In these bins the improvements are larger than the uncertainty margins, and therefore significant. In the coastal and escarpment zone of mainly West Antarctica (the 250–1750 bins), a small overestimation of SMB is now suggested, but the bias is not significantly different from zero.

### 3.5.2 Snow accumulation radar

Table 3 compares simulated annual SMB with annual accumulation rates derived from the snow radar in three regions in West Antarctica (Fig. 1). This comparison includes two new but small (unpublished) surveys over Getz and Ronne ice shelves. Correlations for annual values between the measured and simulated SMB have significantly improved for the three surveys, clearly resulting from the better constrained interannual variability (Van de Berg and Medley, 2016). For the largest of the survey areas, Thwaites glacier, correlation r has improved from 0.68 to 0.90, comparable to the reanalyses in this region. Modelled mean accumulation rates over Thwaites remain slightly underestimated, but changes are small.

For the ice shelf comparisons correlation (r) has increased significantly from $\sim$0.5 to $\sim$0.8. These values are lower than the Thwaites comparison because (1) the accumulation rate over the Ronne ice shelf survey is lower ($\sim$190 mm w.e. y$^{-1}$) resulting in a lower signal-to-noise ratio in the radar time series and (2) the Getz record is shorter.

### 3.5.3 GRACE

Figure 9 shows detrended and deaccelerated mass anomalies from GRACE, RACMO2.3p2 and RACMO2.3p1 for the West and East Antarctica ice sheets, the AP and for the AIS. Changes from RACMO2.3p1 to RACMO2.3p2 are small and likely insignificant as changes fall within the uncertainties of both products, but generally there is a slight improvement. For West Antarctica (Figure 9a) and East Antartica (Figure 9c) the RMSD and correlation have both improved slightly. For the AP





changes are smaller and not significant. When these regions are combined, we find a slight improved in the mass anomalies for the whole Antarctic ice sheet, but our main conclusion is that both model versions realistically simulate the Antarctic mass anomalies.

### 3.5.4 Surface melt

To assess modelled surface melt fluxes, Figure 10 correlates annual average (2000–2009) melt fluxes from RACMO2.3 and the QSCAT satellite for the whole AIS and six regions. The increase in snowmelt for the total AIS (inset) is evident and the negative bias has decreased from –61 to –15 Gt y$^{-1}$, while correlation r$^2$ has increased from 0.75 to 0.81. For the six regions, melt is also better represented, although a few years remain in which melt is underestimated.

For the two regions in the AP, Larsen C and Wilkins ice shelves, high-resolution estimates from RACMO2.3/AP show marginal differences with the coarser model version. The largest underestimation is shown for Wilkins ice shelf in both model versions. Here, observed melt fluxes may be overestimated, due to extensive melt ponding and/or saturated firn conditions in this region (Trusel et al., 2013; Välisuo et al., 2014), a feature which negatively affects the QSCAT retrievals.

Figure 11 compares cumulative melt for 1990–2015 by RACMO2.3/ANT, and the melt calculated by an EBM (with two roughness length settings) forced with meteorological data from Neumayer station (Reijmer and Hock, 2008; König-Langlo, 2013). Updated model melt correlates best with the melt calculated by the EBM, but the total melt remains somewhat underestimated, consistent with the comparisons with QSCAT. The timing of the melt events is captured well: many of the strong melt summers e.g. 1994/1995, 2000/2001 and 2013/2014 are now accurately modelled.

## 4 Results: the Antarctic Peninsula at 5.5 km

### 4.1 Changes in modelled SMB and SEB

For RACMO2.3/AP at 5.5 km resolution, changes in SMB and SEB components are mostly consistent with those at the coarser resolution (Fig. 3). This means that for the AP we see: an increase in P$_{tot}$ (Fig. 3a) and hence the SMB (Fig. 3d) over the western slopes and a small decrease in P$_{tot}$ over the eastern ice shelves, a decrease in sublimation (Fig. 3b), and a considerable increase in snowmelt (Fig. 3d). There are only slight local differences in topography related variables such as precipitation, due to the different interpolation setting used to aggregate the model topography, shifting topographic features approximately one grid box northwards. However, as we will show in the next section, this hardly affects integrated SMB estimates.

Table 4 shows the integrated changes for RACMO2.3/AP. Total integrated SMB (1979–2014) has increased by 16 Gt y$^{-1}$ to 366 ± 58 Gt y$^{-1}$. This increase is mainly caused by changes in precipitation (+14 Gt y$^{-1}$). Even though there are large local changes due to re-projection of the RACMO2.3/AP topography, it hardly affects the total integrated SMB. Changes in SU$_{ds}$ and the resulting SU are similar to those at the coarser resolution. While SU$_{ds}$ dropped by 4 Gt y$^{-1}$(44%), SU$_s$ has increased by 1 Gt y$^{-1}$ and total sublimation SU now amounts to 8± 2 Gt y$^{-1}$, a decrease of 3 Gt y$^{-1}$. Snowmelt has increased significantly (+10 Gt y$^{-1}$) but most of the meltwater increase is refrozen in the snowpack (+9 Gt y$^{-1}$). As a result runoff fluxes remain





similar to the fluxes simulated in the old version. Relatively, most of the SMB increase comes from an increase in the eastern Antarctic Peninsula (EAP) SMB (+5.3%), while the SMB in the western Antarctic Peninsula (WAP) has increased by 3.9%.

## 4.2 Evaluation of modelled SMB

Changes in RACMO2.3/AP are generally small and most changes are also visible at the coarser resolution, and only one
previous evaluation with new observational data is revisited.

### 4.2.1 Discharge estimates

Figure 12 shows RACMO2.3/AP average SMB (1979–2016) as a function of glaciers draining in Larsen B (a, Wuite et al., 2015) and George VI ice shelves (b, Hogg et al., 2017) for 1995/1996, when these were assumed to be in balance. Changes due to the model update are small yet different for the two regions. For Larsen B a consistent improvement in modelled SMB is
seen: r$^2$, bias and RMSD have all marginally improved. For George VI SMB values are an order of magnitude larger. The bias has improved for the glaciers where the SMB was formerly underestimated, but worsened for those where it was overestimated. RMSD has worsened slightly, but the slope of the fit and the correlation coefficient $r^2$ have improved. Overall changes are small and fall well within the uncertainty/interannual variability of the discharge estimates/modelled SMB.

## 5 Remaining limitations and challenges

### 5.1 Clouds

In this update of RACMO2.3 the near-surface climate, in terms of the SEB and SMB, has generally improved, but several limitations and challenges remain. For instance, changes in cloud scheme parameters governing the conversion of clouds into precipitation have altered the distribution and quantity of clouds over the ice sheet, which in turn has led to improvements in the near-surface climate, as presented in this study. Several of the variables related to the distribution of clouds over the East
Antarctic plateau (LW$_d$, precipitation and surface temperature T$_s$), are now overestimated in the interior, albeit slightly so. Therefore, further improving these biases will require a different approach.

For instance, when the model is used at increasingly higher spatial resolution, it becomes important whether precipitation is handled prognostically by the model: while it is now assumed that precipitation reaches the surface instantaneously, ideally, precipitation should be modelled as a prognostic variable in order to capture its fall time and horizontal displacement. This
could significantly affect local precipitation patterns and hence SMB in regions of relatively high wind speeds and snowfall rates, such as the AP and the WAIS.

Several studies suggest that not necessarily cloud amount, but the fractionation of cloud ice and cloud water content is important for the radiative fluxes: King et al. (2015) showed that significant biases exist in relation to the ice/water content of clouds, and that a realistic simulation mainly requires accurate modelling of the presence or absence of clouds at a particular level
in the atmosphere. Matus and L'Ecuyer (2017); Lenaerts et al. (2017) show that climate models typically exhibit significant



biases in the simulation of clouds and radiation in the polar regions, and that future model developments should focus on the microphysical properties of clouds, and their radiative impact. We have also shown in Figure 6 that, even though the simulation of the radiative fluxes has improved at the surface, RACMO2.3p2 considerably overestimates cloud ice and cloud water, suggesting that compensating errors exist due to a cloud radiation scheme that is not active enough. Further model improvement

efforts should also focus on the surface layer turbulent mixing scheme (Van de Berg et al., 2006).

### 5.2    Cold bias, meltwater and hydrostatic assumption

The updated RACMO2.3/ANT and RACMO2.3/AP remain somewhat too cold at the surface, especially at lower elevations. This is partly due to biases in the SEB and cloud cover, but also likely due to the limited horizontal resolution. A limited horizontal resolution affects the near-surface winds, where biases influence the simulation of SHF, and the associated mixing

of warmer air to the surface. This limitation is addressed by the higher resolution in RACMO2.3/AP, but a further increase of the horizontal resolution moves the model towards the limits of the hydrostatic assumption, and it would become necessary to resolve vertical accelerations and internal and gravity waves. A non-hydrostatic model formulation should be used if these features are to be explored further.

The cold bias in the model 10 m firn temperature mostly affects more temperate regions of the ice sheet (Fig. 5), which

suggests that the bias could be related to the refreezing of meltwater. Meltwater percolates into the snow column, refreezes and raises snow temperature. Underestimated surface melt, or neglecting meltwater penetration features (Harper et al., 2012; Ligtenberg et al., 2014) could result in too low snow temperature. Another explanation is improper modelling of the albedo-melt feedback in regions of sustained local melt, such as the melt induced by foehn winds in the AP (Luckman et al., 2014; Van Wessem et al., 2016), or the wind-albedo interaction over blue ice areas in DML (Lenaerts et al., 2016b). Further efforts

to improve modelled melt fluxes should therefore focus on these features.

### 6    Summary and conclusions

In this study we evaluated the modelled Antarctic ice sheet (AIS) climate, surface mass balance (SMB) and surface energy balance (SEB) in RACMO2.3p2, the latest polar version of the regional atmospheric climate model RACMO2 (1979–2016). This model version is applied at 27 km horizontal resolution to the full AIS, and at 5.5 km resolution to the Antarctic Peninsula

(AP). Updates to the previous model version include additional upper-air relaxation by ERA-Interim re-analyses data, a revised topography, tuned parameters in the cloud scheme in order to generate more precipitation towards the AIS interior, and modified snow properties reducing drifting snow sublimation and increasing simulated surface snowmelt. The updates in the 5.5 km AP simulation are similar, but do not include a new topography or upper-air relaxation.

The updated model simulates more clouds towards the interior of the ice sheet, increasing downwelling longwave radiation

and snowfall rates. Drifting snow sublimation rates have decreased by 43% and surface snowmelt has doubled. In total, these changes have led to an integrated SMB for the ice sheet including ice shelves and excluding the AP, of 2229 Gt y$^{-1}$, with an interannual variability $\sigma = 109$ Gt y$^{-1}$. We evaluated the model with various observational data, including in-situ SMB




observations (Favier et al., 2013), radar-derived snow accumulation (Medley et al., 2013), Cloudsat-CALIPSO satellite observations (Van Tricht et al., 2016b), QSCAT-satellite derived melt fluxes (Trusel et al., 2013), AP glacial discharge estimates (Wuite et al., 2015; Hogg et al., 2017) and SEB data from AWSs. We find a significant improvement in simulated SEB and SMB over the ice sheet, in particular at lower elevations. The largest improvement is found in modelled surface snowmelt, that

now compares considerably better with the QSCAT and Neumayer (indirect) melt observations. No significant changes are found for the 5.5 km model version: here, model results are comparable to earlier versions.

This study shows that the latest version of RACMO2 can be used for high-resolution future projections of AIS SMB and SEB. However, limitations remain, mostly related to the cloud microphysics, the horizontal resolution, and party to the meltwater percolation scheme in the snow model.

**7   Data availability**

The RACMO2.3p2/ANT (27 km) and RACMO2.3p2/AP (5.5 km) data from 1979–2016 presented in this study are available from the authors without conditions.

*Author contributions* J. M. W., B. P. Y. N., W. J. B. and M. B. conceived this study, decided on the new model setting and performed the analysis and synthesis of the data sets. J. M. W. performed the model simulations, led the writing of the manuscript. G. B., C. L. J., K. K. S. L., B. M, C. H .R, K. T., L. D. T., B. W. and J. W. processed and provided observational data sets. J. M. W., B. P. Y. N, W. J. B, M. B., J. T. M. L, S. R. M. L, C. L. J., C. H. R., E. M., and L. U. contributed to the development of the model. All authors contributed to discussions in writing this manuscript.

*Competing interests* The authors declare that they have no conflict of interest.

*Acknowledgements.* We acknowledge financial contributions made by the Netherlands Organization for Scientific Research (grant 866.15.201) and the Netherlands Earth System Science Center (NESSC). We thank the ECMWF for the use of their supercomputing facilities. Graphics and calculations were made using the NCAR Command Language (Version 6.3.0, 2017).



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



**Table 1.** The AWS topographic characteristics and period of operation (until December 2011). Nr. months represents the number of available months over total months (725/770) of the model period (January 1979–December 2011). If no end time is indicated, the AWS is still operational end of 2011. For Neumayer station only cumulative melt is used in this study (Sect. 3.5.4.)

| AWS | 4 | 5 | 6 | 8 | 9 | 10 | 11 | 12 | 16 | Neumayer |
|---|---|---|---|---|---|---|---|---|---|---|
| Latitude | 72°45′ S | 73°06′ S | 74°28′ S | 76°00′ S | 75°00′ S | 79°34′ S | 71°09′ S | 78°39′ S | 71°57′ S | 70°39′ S |
| Longitude | 15°29.9′ W | 13°09.9′ W | 11°31.0′ W | 08°03′ W | 00°00′ E/W | 45°47′ W | 06°42′ W | 35°38′ E | 23°20′ E | 8°15′ W |
| Elevation (obs) | 34 m | 363 m | 1160 m | 2400 m | 2892 m | 890 m | 700 m | 3620 m | 1300 m | 40 m |
| Elevation (mod) | 23 m | 332 m | 1219 m | 2405 m | 2856 m | 789 m | 224 m | 3621 m | 1130 m | 55 |
| Start | Dec 1997 | Feb 1998 | Jan 1998 | Jan 1998 | Dec 1997 | Jan 2001 | Jan 2007 | Dec 2007 | Feb 2009 | Jan 1992 |
| End | Dec 2002 | – | Jan 2009 | Jan 2003 | – | Jan 2006 | – | – | – | Dec 2015 |
| nr. months | 60/60 | 167/167 | 134/134 | 19/44 | 162/168 | 48/54 | 57/59 | 49/49 | 29/35 | |

**Table 2.** Total ice sheet, including ice shelves and excluding the Antarctic Peninsula, integrated SMB mean 1979–2014 (the overlapping period for both model versions) values [Gt y$^{-1}$] with interannual variability $\sigma$: total (snow+rain) precipitation (P$_{tot}$), snowfall (SN), rainfall (RA), total sublimation (SU$_{tot}$), surface sublimation (SU$_s$), drifting snow sublimation (SU$_{ds}$), drifting snow erosion (ER$_{ds}$), runoff (RU), snowmelt (M) and refrozen mass (RF). All values calculated on the RACMO2.3p2/ANT ice-mask (Bamber and Gomez-Dans, 2009). EAIS, WAIS and ISLANDS SMB are based on the Rignot et al., 2017 (in preparation) drainage basins, and denote the grounded ice sheet. No values for RACMO2.3p1/ANT are given for the latter region; as they don't correspond with the RACMO2.3p1 ice mask used.

| | RACMO2.3p2 mean | $\sigma$ | RACMO2.3p1 mean | $\sigma$ | p2 – p1 mean |
|---|---|---|---|---|---|
| P$_{tot}$ | 2396 | 110 | 2386 | 118 | +10 (0.5%) |
| SN | 2394 | 110 | 2383 | 109 | +11 (0.5%) |
| RA | 3 | 1 | 2 | 1 | 1 (50%) |
| SU$_{tot}$ | 161 | 7 | 217 | 11 | –56 (25%) |
| SU$_s$ | 59 | 4 | 37 | 3 | +22 (60%) |
| SU$_{ds}$ | 102 | 5 | 181 | 9 | –79 (43%) |
| ER$_{ds}$ | 5 | 0.5 | 5 | 0.5 | 0 |
| RU | 3 | 1 | 3 | 1 | 0 |
| M | 71 | 28 | 36 | 17 | +35 (97%) |
| RF | 71 | 28 | 36 | 17 | +35 (97%) |
| SMB (TotIS) | 2229 | 109 | 2160 | 118 | +69 (3.2%) |
| SMB (GIS) | 1885 | 95 | 1782 | 103 | +103 (5.8%) |
| SMB (EAIS) | 1130 | 80 | 1051 | 94 | +79 (7.5%) |
| SMB (WAIS) | 644 | 63 | 627 | 60 | +17 (2.7%) |
| SMB (ISLANDS) | 110 | 11 | – | – | – |

| | Thwaites | Getz | Ronne |
|---|---|---|---|
| Radar coverage | ∼1650 km | ∼75 km | ∼50 km |
| Snow radar SMB | 0.457 | 0.897 | 0.186 |
| RACMO2.3p1 SMB | 0.447 | 0.976 | 0.189 |
| RACMO2.3p2 SMB | 0.438 | 0.942 | 0.158 |
| | Correlation (r) with radar | | |
| RACMO2.3p1 | **0.68** | 0.49 | **0.51** |
| RACMO2.3p2 | **0.90** | **0.79** | **0.80** |

**Table 3.** Comparison of RACMO2.3p2 and RACMO2.3p1 SMB (mm w.e. y$^{-1}$) with radar derived snow accumulation (Medley et al., 2013) for three regions in West Antarctica: Thwaites glacier (mean latitude/longitude: 77.80 °S,104.66 °W) and Getz (74.51°S, 125.54°W) and Ronne (80.39°S, 72.21 °W) ice shelves. Significant (>95%) correlations are denoted in bold.





**Figure 1.** Map of Antarctica showing the in-situ SMB observations (red dots), AWSs (blue diamonds), the three snow radar regions (green dots), 64 10 m snow temperature observations (black dots) used in this study. Shown are the relaxation zones for both RACMO2.3p2/ANT and RACMO2.3p2/AP with the black dotted rectangles, as well as the RACMO2.3p1/AP domain (blue rectangle for the non-overlapping boundary). Also shown are the grounding line in black and relevant locations as used throughout the text (with ice shelves in italics).



**Figure 2.** Annual average (1979–2014) difference (RACMO2.3p2 – RACMO2.3p1) of (a) downwelling longwave radiation ($LW_d$), (b) surface temperature ($T_s$), (c) 10 m wind speed ($V_{10m}$) and (d) sensible heat flux (SHF). Areas where the difference exceeds one unit of standard deviation of the difference are stippled.





Difference RACMO2.3p2 - RACMO2.3p1 for 1979-2014

**Figure 3.** Annual average (1979–2014) difference (RACMO2.3p2 – RACMO2.3p1) of (a) total precipitation (snow + rain), (b) total drifting (snow + surface) sublimation, (c) snowmelt and (d) SMB. Areas where the difference exceeds one unit of standard deviation of the difference are stippled.





**Figure 4.** Correlation plots of RACMO2.3p2 (blue) and RACMO2.3p1 (red) as a function of the observation for (a) sensible heat flux (SHF), (b) latent heat flux (LHF), (c) $SW_{net}$, (d) $LW_{net}$, (e) $V_{10m}$ and (f) $T_s$. Denoted for all variables are the slope Rc), correlation coefficient $r^2$, bias and root mean square deviation (RMSD) for both model versions.





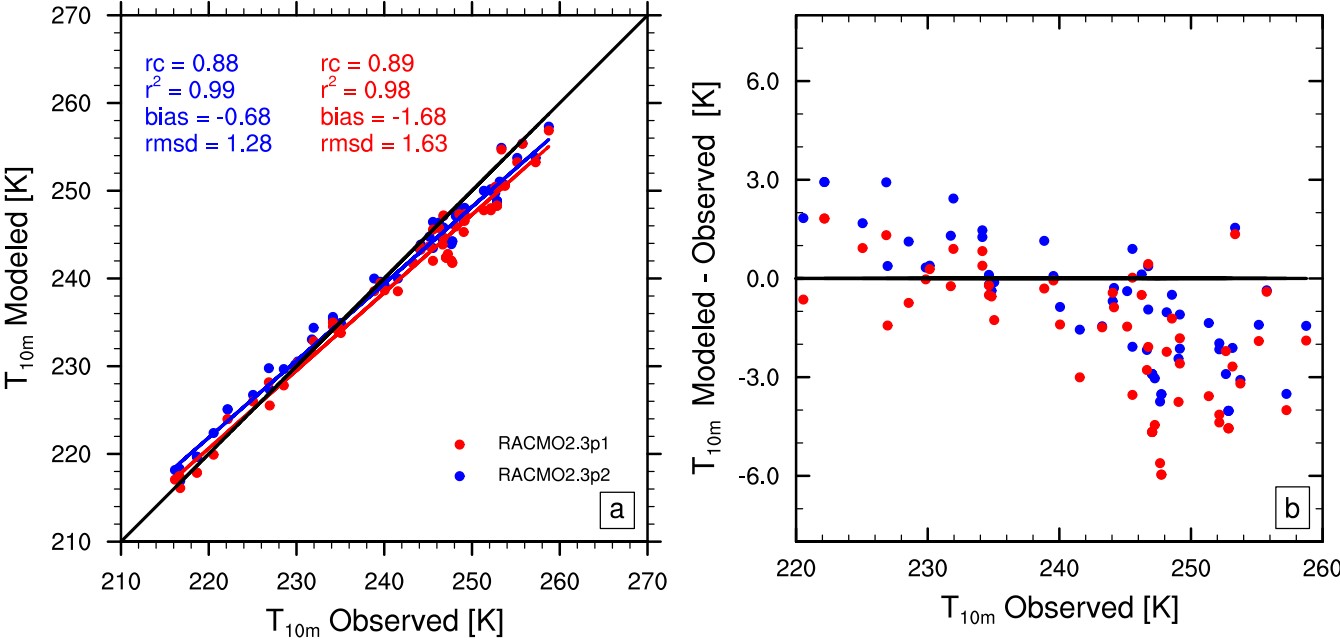

**Figure 5.** (a) RACMO2.3p2 (blue) and RACMO2.3p1 (red) and (b) modelled – observed 10 m snow temperature as a function of the 10 m snow temperature observations. Denoted are the slope, correlation coefficient $r^2$, bias and root mean square deviation (RMSD) for both model versions.

**Table 4.** Total Antarctic Peninsula integrated SMB mean 1979–2014 (the overlapping period for both model versions) values [Gt y$^{-1}$] with interannual variability $\sigma$: total (snow+rain) precipitation ($P_{tot}$), snowfall (SN), rainfall (RA), total sublimation ($SU_{tot}$), surface sublimation ($SU_s$), drifting snow sublimation ($SU_{ds}$), runoff (RU), snowmelt (M) and refrozen mass (RF). All values calculated on the RACMO2.3p2/AP ice-mask without Larsen B included.

|  | RACMO2.3p2 | | RACMO2.3p1 | | p2 –p1 |
|---|---|---|---|---|---|
|  | mean | $\sigma$ | mean | $\sigma$ | mean |
| $P_{tot}$ | 377 | 58 | 363 | 57 | +14 (0.4%) |
| SN | 374 | 57 | 360 | 56 | +14 (0.4%) |
| RA | 3 | 1 | 3 | 1 | 0 |
| $SU_{tot}$ | 8 | 2 | 11 | 2 | −3 (27%) |
| $SU_s$ | 3 | 0.5 | 2 | 1 | +1 (5%) |
| $SU_{ds}$ | 5 | 0.5 | 9 | 2 | −4 (44%) |
| RU | 1 | 2 | 2 | 2 | 0 |
| M | 41 | 16 | 31 | 12 | +10 (32%) |
| RF | 43 | 14 | 33 | 12 | +8 (24%) |
| SMB (AP) | 366 | 58 | 350 | 57 | +16 (4.5%) |
| SMB (WAP) | 287 | 48 | 276 | 47 | +11 (3.9%) |
| SMB (EAP) | 79 | 10 | 75 | 11 | +4 (5.3%) |





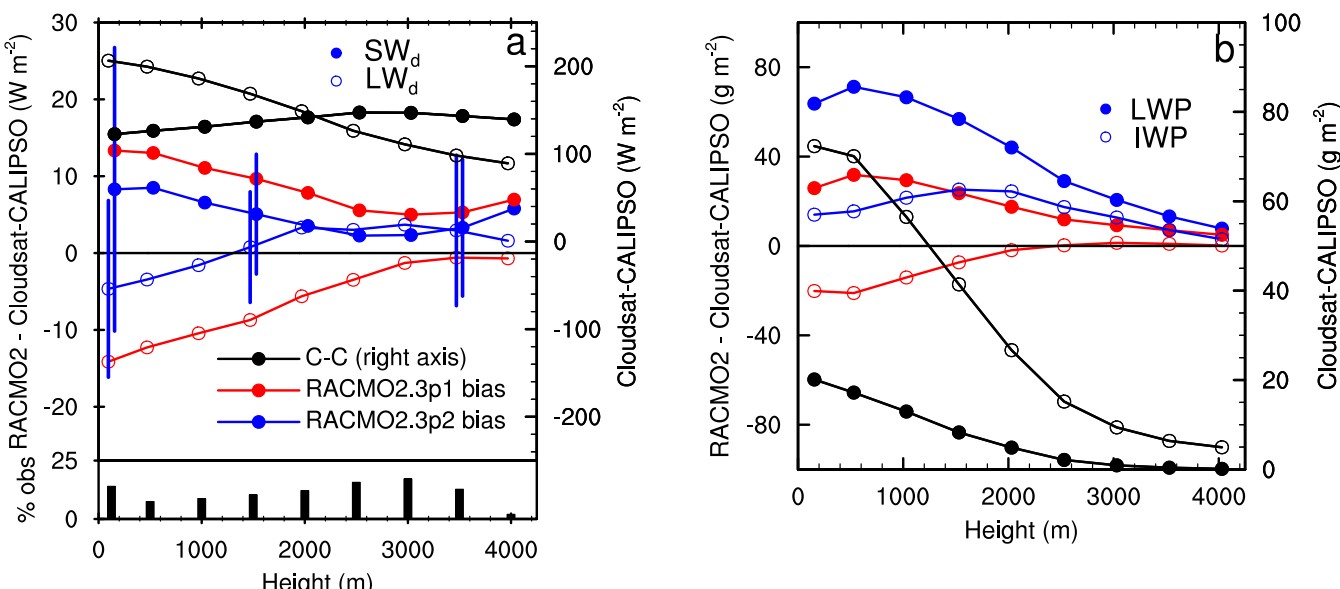

**Figure 6.** (a) Absolute bias (left axis) in average (2007–2010) modelled downwelling longwave radiation (LW$_d$) (open circles) and down-welling shortwave radiation (SW$_d$) (closed circles) and (b) cloud ice water path (IWP) (open circles) and cloud liquid water path (LWP) (closed circles) for RACMO2.3p2 (blue) and RACMO2.3p1 (red) compared to the Cloudsat-CALIPSO product (Van Tricht et al., 2016b). Also shown on (a) and (b) are the observational data (black circles, right axis). Error bars in (a) de-note the root mean square difference (RMSD) from a statistical comparison of the Cloudsat-CALIPSO fluxes and ground-based AWS observations (Van Tricht et al., 2016a), only shown for three bins as based on the elevations of the four ground-based AWSs used. The data are binned in 500 m surface elevation intervals (0–250, 250–750, etc). The bar chart in (a) denotes the percentage of total gridpoints per elevation bin. To separate LW$_d$ and SW$_d$ fluxes, x-axis locations of each bin in (a) are displaced by 60 m.



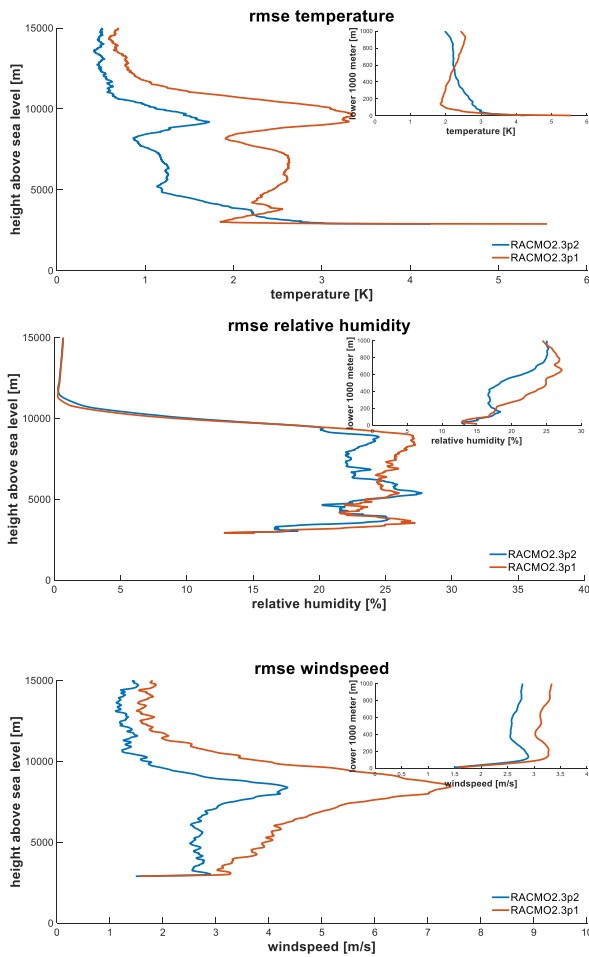

**Figure 7.** Root mean square error (RMSE) of average upper atmosphere temperature (a), relative humidity (b) and wind speed (c) from RACMO2.3p2 (blue) RACMO2.3p1 (red) and the radiosonde observations at Kohnen station, from January 2014. Note that the lowermost level is at the surface of Kohnen station, at ∼ 3000 m elevation. Subpanels show a zoom of the lower 1000 m.





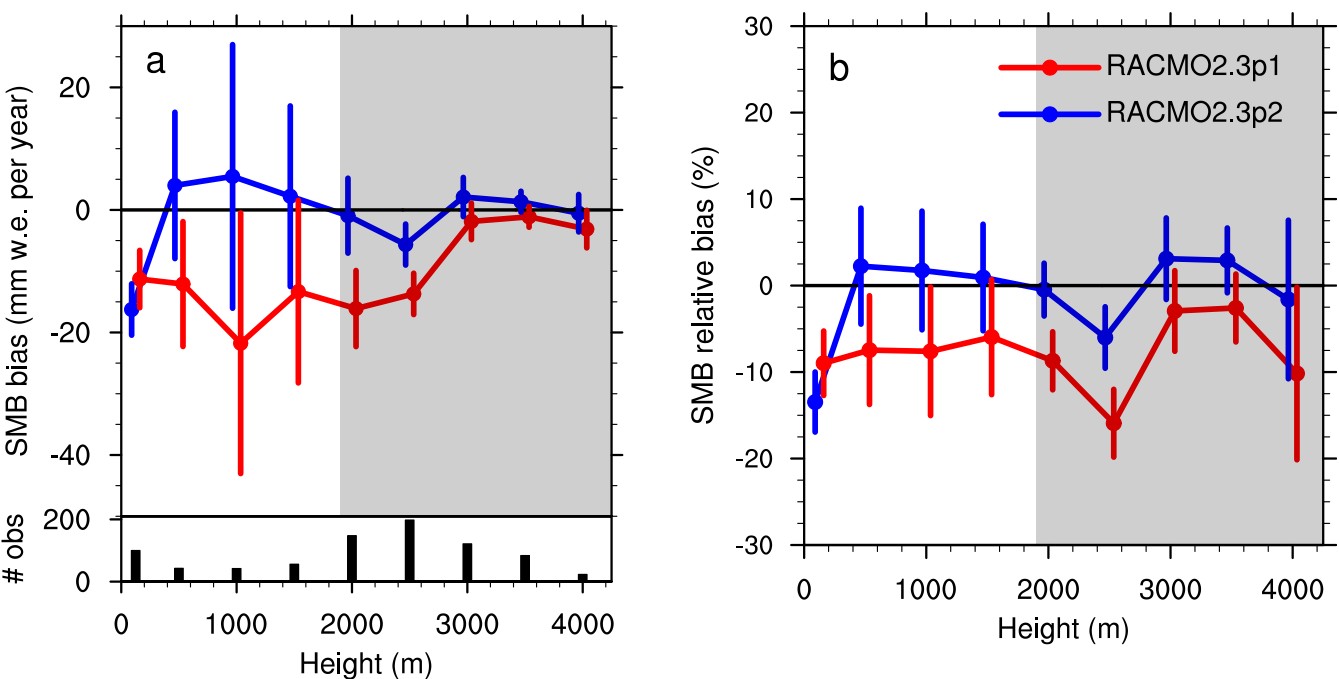

**Figure 8.** (a) Absolute bias and (b) relative bias ((model – observation)/model x 100%) in modelled SMB for RACMO2.3p2 (blue) and RACMO2.3p1 (red). The data are binned in 500 m surface elevation intervals (0–250, 250–750, etc.). Error bars denote the combined uncertainty of the model and observations within each height bin, based on Van de Berg et al. (2006); Van Wessem et al. (2014a). Elevations above 2000 m (shaded grey) represent East Antarctica exclusively. The bar chart in (a) denotes the amount of weighted observations in each bin. To separate blue and red lines, x-axis locations of each bin are displaced by 75 m.





**Figure 9.** Detrended and deaccelerated average (for details see text) SMB anomalies of RACMO2.3p2 (blue), RACMO2.3p1(red) and GRACE (black), release RL05, for West Antarctica (a), the Antarctic Peninsula (b), East Antarctica (c) and the whole Antarctic ice sheet (AIS, d). Shown are root mean square deviation (RMSD) and correlation ($r^2$).





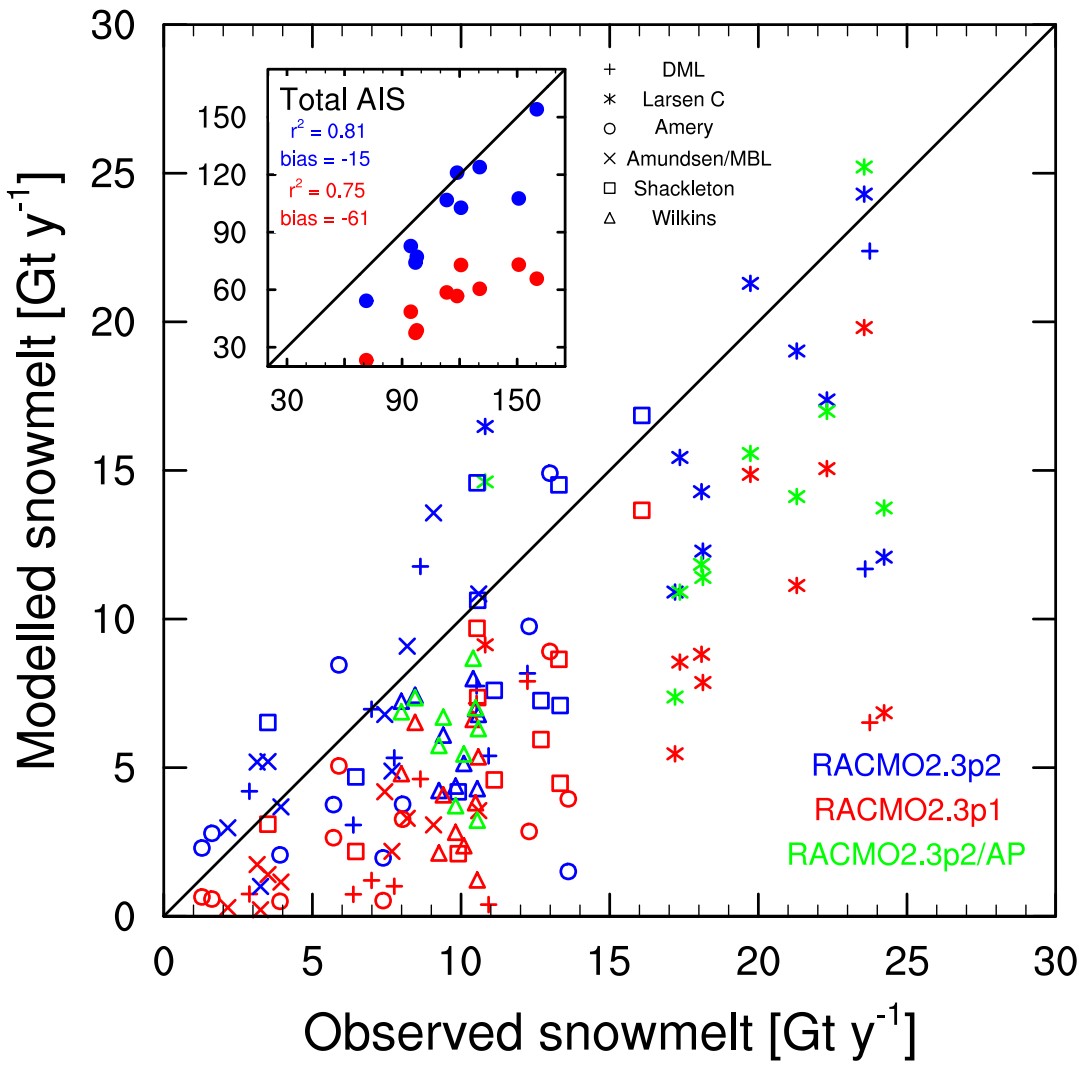

**Figure 10.** Modelled RACMO2.3p2 (blue), RACMO2.3p1 (red) and RACMO2.3p2/AP (green) as a function of observed QSCAT (Trusel et al., 2013) snowmelt in $\mathrm{Gt}\,\mathrm{y}^{-1}$. Shown are yearly 2000-2009 values for the total AIS (subpanel) and area averaged values for six regions in Antarctica defined in Trusel et al. (2013): Dronning Maud Land (DML, plus signs), Larsen C ice shelf (asterisks), Amery Ice Shelf (open circles), Amundsen/Marie Byrd Land (crosses), Shackleton Shelf (squares) and Wilkins Ice Shelf (triangles). Denoted are correlation $r^2$ and bias for the total AIS.



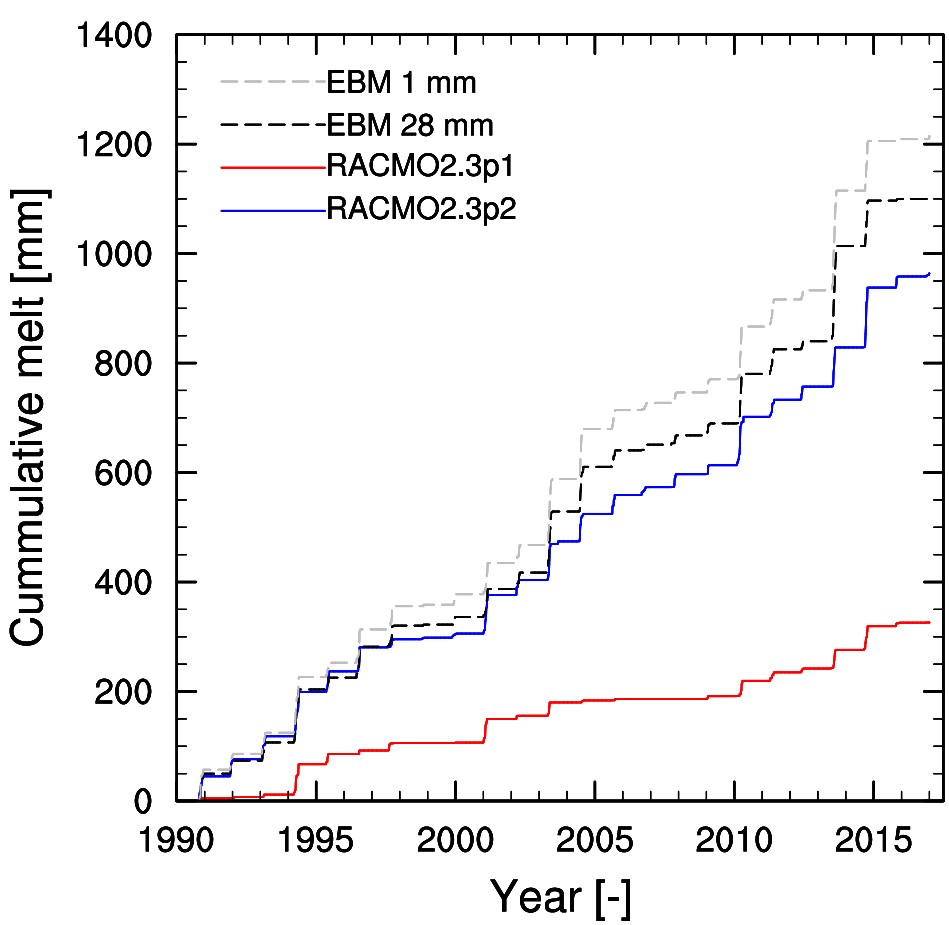

**Figure 11.** Cumulative melt at Neumayer calculated by the energy balance model (EBM) for 1990–2015 with roughness length $z_0 = 28$ mm (dashed black line), $z_0 = 1$ mm (dashed grey line), and as modelled by RACMO2.3p2 (blue) and RACMO2.3p1 (red).





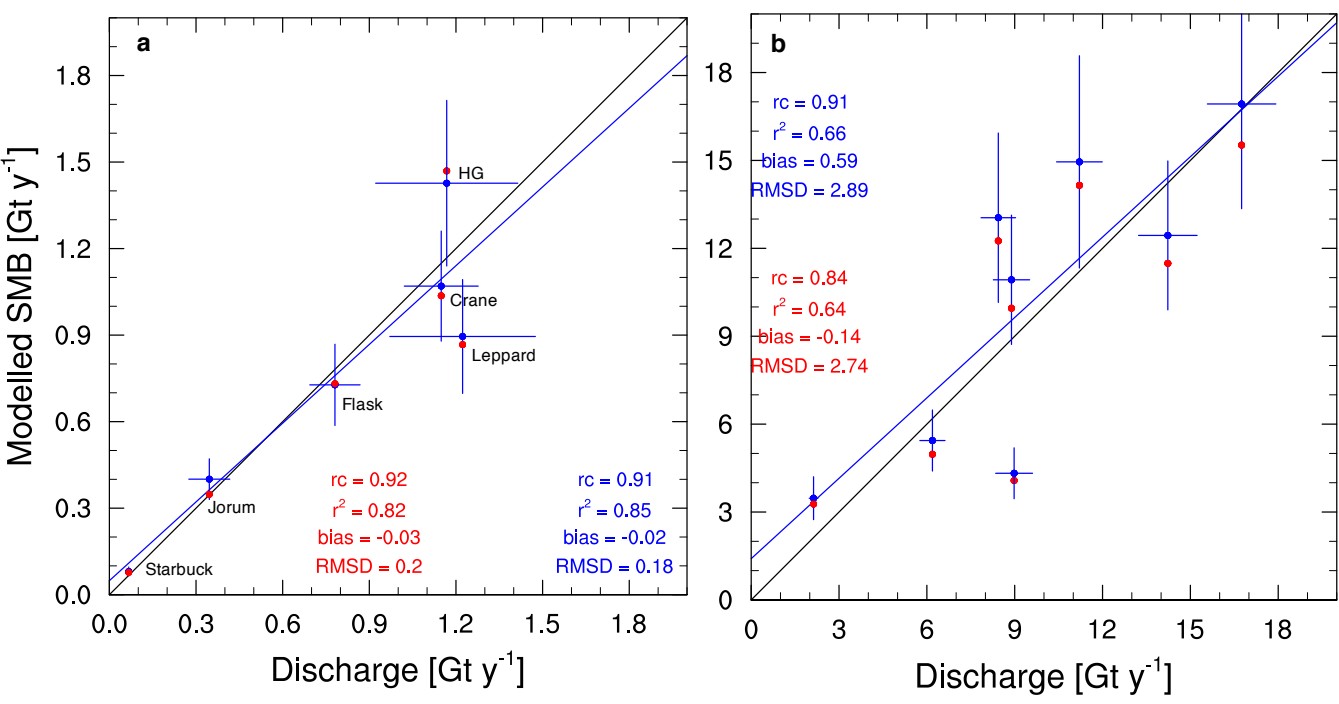

**Figure 12.** Modelled RACMO2.3p2/AP (blue) and RACMO2.3p1/AP (red) integrated average (1979–2016) SMB as a function of glacial discharge estimates from (a) the Larsen B embayment (Wuite et al., 2015) and (b) the George VI embayment (Hogg et al., 2017). Horizontal error bars represent the uncertainty in the discharge estimate, vertical bars the interannual variability of RACMO2.3.