# Peer review of "Modelling the climate and surface mass balance of polar ice sheets using RACMO2, part 2: Antarctica (1979–2016)."

_The Cryosphere, 2017_

## Referee Comment (RC1) · M. Lehning (Referee) · 6 Dec 2017

The updated simulations presented in this paper are interesting and worth publishing. While in general, I would not support that a new paper is written after simply adjusting some model parameters, this paper has value because it clearly indicates, where current models need to be improved and where processes such as cloud dynamics are incompletely understood. The numbers presented are the current state of the art and the basis for other researchers working on Antarctic mass balance questions. Nonetheless, I would like to make some major suggestions to implement before publication. What is lacking in the paper is an in-depth discussion of the generality or universality

of model assumptions and choices. For example, if RACMO were to be run over the Himalayas with the updated cloud parameterization, what would happen there? Also, the critical model parts need to be better described such that a self-contained paper results. In particular:

1) The main adjustment made to the critical cloud water and cloud ice content parameters appears massive and is not discussed in terms of what it means. While I can see that a full presentation of the cloud model is beyond the scope of the effort, to present the main features of the cloud model and the role of the parameters should become an (adequate) part of the methodological description.

2) Since some important improvements in the model results are caused by the newly introduced upper air relaxation process, this should also be introduced in a methodology section. In particular, if there are any interactions to expect from this assimilation with the cloud parameter changes, this should be discussed in detail.

3) A similar comment applies to the discussion of the drifting snow parameterization. While strict validation of the model is difficult, a more in-depth discussion of available measurements and model results is required. In particular, I would request that comparisons are made to the (limited but existing) Antarctic snow transport results from other groups (e.g. Amory et al., 2015 and references therein). It could be also helpful to discuss similar efforts made in Alpine terrain (Vionnet et al., 2014; Zwaaftink et al., 2013) and to comment on potential errors in the parameterization of drifting snow sublimation in light of newer results (e.g. Dai and Huang, 2014).

4) One other major request is related to data publishing: While the authors say that the model results are available, the observational data should also be published in the form they have been used for the validation. The value of the paper relies heavily on the comprehensive data collection the authors have been able to accomplish. It is very valuable for the scientific community to get these comprehensive validation data along with the paper. Otherwise the data collection effort has to be repeated by authors who

want to do similar validation work but maybe with different models. I understand that some data sets may not be available for publication if the permission of the original data collector is not given. But I would expect that this is not the case for a majority of the data used for validation and thus those data sets can be made available on a common data platform (e.g. Pangea, NSIDC). Some of the data sets have been processed by the authors (e.g. the accumulation radar data if I understand correctly) and would not be available to the scientific community if not published along with the paper.

Detailed Comments: In general, form of presentation and language are very good and the paper reads very well. A few smaller things are noted below.

Figure 5 and corresponding discussion: Looks like the improvement in bias and RMSE is only because a few rather warm temperatures have been simulated better but this in a temperature range, in which the data show a lot of scatter. However, the new simulations clearly show a larger bias at low temperatures and this may be worth mentioning?

Can you give a reason why you do this elevation binning in Figure 8 instead of showing a map?

Section 3.5.4: Appears that you already start the comparison between coarser and higher-resolution model runs here; somewhat confusing since this is revisited in Section 4?

References: Amory, C., A. Trouvilliez, H. Gallee, V. Favier, F. Naaim-Bouvet, C. Genthon, C. Agosta, L. Piard, and H. Bellot (2015), Comparison between observed and simulated aeolian snow mass fluxes in Adelie Land, East Antarctica, Cryosphere, 9(4), 1373-1383, doi:10.5194/tc-9-1373-2015.

Dai, X., and N. Huang (2014), Numerical simulation of drifting snow sublimation in the saltation layer., Scientific reports, 4, 6611, doi:10.1038/srep06611.

Vionnet, V., E. Martin, V. Masson, G. Guyomarc'H, F. Naaim-Bouvet, A. Prokop, Y. Durand, and C. Lac (2014), Simulation of wind-induced snow transport and sublimation

in alpine terrain using a fully coupled snowpack/atmosphere model, Cryosphere, 8(2), 395–415, doi:10.5194/tc-8-395-2014.

Zwaaftink, C. D. G., R. Mott, and M. Lehning (2013), Seasonal simulation of drifting snow sublimation in Alpine terrain, Water Resources Research, 49(3), 1581-1590, doi:10.1002/wrcr.20137.

---

## Referee Comment (RC2) · Anonymous Referee #2 · 7 Dec 2017

This is a high-quality comprehensive evaluation of the revised version of RACMO2.3 model over Antarctica in relation to its prior formulation for a wide variety of variables. The use of AWS and upper air data is limited however. Overall the improvements are modest but most notable in the surface mass balance components. There are a small number of issues.

1. Page 2, line 2: Many, many authors have discussed expected Antarctic warming is accompanied by increased snow fall estimates, so please add 1-2 more references. One recent example is Palerme et al. (2017) on CMIP5 models in Climate Dynamics. 2. Page 2, Line 26: Drop "so-called". 3. Page 2, Line 47: The atmosphere does not

interact with topography rather its motion is impacted by topography. 4. Page 2, Line 63: airborne elevation data - do you mean from Icebridge? Otherwise contemporary amounts must be very small. 5. Section 3.4.2: Remind the reader that the 10-m firn temperature closely matches the annual mean 2-m air temperature. By the way, what is Ts? 6. Figure 4: Are these values monthly means? 7. Figure 7: Present the biases as well as the RMSEs. Discuss in the text.

---

## Author Comment (AC1) · 19 Jan 2018

Please find the revised manuscript and responses to both reviewers in the attached file.

Please also note the supplement to this comment:
https://www.the-cryosphere-discuss.net/tc-2017-202/tc-2017-202-AC1-supplement.zip

---

## Author Response (AR1)

Interactive comment on "Modelling the climate and surface mass balance of polar ice sheets using RACMO2, part 2: Antarctica (1979–2016)"

**M. Lehning (Referee)**

RC: The updated simulations presented in this paper are interesting and worth publishing. While in general, I would not support that a new paper is written after simply adjusting some model parameters, this paper has value because it clearly indicates, where current models need to be improved and where processes such as cloud dynamics are incompletely understood. The numbers presented are the current state of the art and the basis for other researchers working on Antarctic mass balance questions. Nonetheless, I would like to make some major suggestions to implement before publication.
AC: We thank the referee for the positively critical review of our manuscript. We will discuss your comments below.

RC: What is lacking in the paper is an in-depth discussion of the generality or universality of model assumptions and choices. For example, if RACMO were to be run over the Himalayas with the updated cloud parameterization, what would happen there?
AC: We have added an explanation of the model update to section 2.3, explaining that the model update incorporates changes that are simultaneously implemented in the model versions used for Greenland and Antarctica, but is not specific for other regions. *"A special effort is done to synchronise the model updates to the Greenland and Antarctica model versions. Most of the updates presented below are therefore implemented for Greenland as well, unless noted otherwise."* Since RACMO2.1, the polar version of this model has only been used for the high-latitude regions. The cloud parameterization update is relatively simple (see below): clouds will precipitate later and higher up in the air. Over the Himalaya's we would therefore expect a similar response as we see in the Antarctic Peninsula: a slight strengthening of orographic precipitation.

RC: Also, the critical model parts need to be better described such that a self-contained paper results. In particular:

1) The main adjustment made to the critical cloud water and cloud ice content parameters appears massive and is not discussed in terms of what it means. While I can see that a full presentation of the cloud model is beyond the scope of the effort, to present the main features of the cloud model and the role of the parameters should become an (adequate) part of the methodological description.

RC: We have added a more detailed explanation in the data/methods section: "*Previous studies found that the previous model version systematically underestimates snowfall and downwelling longwave radiation in the interior of the ice sheets of both Greenland (Noël et al., 2015) and Antarctica (Van Wessem et al., 2014b). Therefore, the critical cloud water and cloud ice content ($l_{crit}$) thresholds governing the onset of effective precipitation formation for mixed-phase and ice clouds, are increased by a factor 2 (Eqs. 5.35 and 6.39 in ECMWF-IFS (2008)) and 5 (Eq. 6.42 in ECMWF-IFS (2008)), respectively. This increase leads to both ice and water clouds to last longer in the atmosphere before precipitating, and therefore to be advected further towards in the ice sheet interior. As a result, we expect increased cloud*

*cover for colder conditions and higher elevations, and precipitation simulated further inland. Consequently, we expect to further decrease SEB and SMB biases in the AIS interior, in a way similar to Van Wessem et al. (2014a, b).*

*The values of $l_{crit}$ adopted in RACMO2 were obtained after conducting a series of sensitivity experiments, i.e. oneyear simulations, over Greenland to test the dependence of precipitation formation efficiency, spatial distribution and cloud moisture content on $l_{crit}$ and other cloud tuning parameters. From these experiments, we found a linear relationship between $l_{crit}$ for mixed and ice clouds, the vertical integrated cloud content. These new settings were then tested over Greenland and Antarctica for a longer period and proved to raise the accumulation and downwelling radiation fluxes in the interior ice sheets Greenland (Noël et al., 2017) and Antarctica (see Sections 3.1, 3.2 and 3.4.3). The new value of $l_{crit}$ remains well within the range of values previously presented in the literature (Lin et al., 1983).*"

RC: 2) Since some important improvements in the model results are caused by the newly introduced upper air relaxation process, this should also be introduced in a methodology section. In particular, if there are any interactions to expect from this assimilation with the cloud parameter changes, this should be discussed in detail.
AC: The effect of the upper air relaxation is extensively explained in Van de Berg et al., 2016, and is cited accordingly in the manuscript. For clarification, we have added a more detailed explanation in the data/methods section:
"*To better simulate SMB interannual variability in RACMO2.3p2, upper atmosphere relaxation (UAR or nudging) of temperature and wind fields is applied every 6 hours for the model atmospheric levels above 600 hPa (Van de Berg and Medley,2016). UAR is not applied to atmospheric humidity fields in order not to alter clouds and precipitation formation in RACMO2. Additional details about the implementation and the effects of upper air relaxation are found in (Van de Berg and Medley, 2016*).

RC: 3) A similar comment applies to the discussion of the drifting snow parameterization. While strict validation of the model is difficult, a more in-depth discussion of available measurements and model results is required. In particular, I would request that comparisons are made to the (limited but existing) Antarctic snow transport results from other groups (e.g. Amory et al., 2015 and references therein). It could be also helpful to discuss similar efforts made in Alpine terrain (Vionnet et al., 2014; Zwaaftink et al., 2013) and to comment on potential errors in the parameterization of drifting snow sublimation in light of newer results (e.g. Dai and Huang, 2014).
AC: We have added Sections 2.5.5 and 3.5.4 and a Figure (see Figure 1) addressing the snow drift flux results, comparing these to data from 2013 discussed in Amory et al., 2017. These are drifting snow *transport* measurements, and do not enable a comparison with either sublimation or erosion of drifting snow, which are both hard to measure. However, they do provide insight on how much the linear saltation snow load parameter, which is directly related to how much snow is being transported, has improved the results. We thank the reviewer for this suggestion, as the addition of this dataset has surely improved the manuscript.

RC: 4) One other major request is related to data publishing: While the authors say that the model results are available, the observational data should also be published in the form they have been used for the validation. The value of the paper relies heavily on the comprehensive data collection the authors have been able to accomplish. It is very valuable for the scientific community to get these comprehensive validation data along with the paper. Otherwise the data collection effort has to be repeated by authors who want to do similar validation work but maybe with different models. I understand that some data sets may not be available for publication if the permission of the original data collector is not given. But I would expect that this is not the case for a majority of the data used for validation and thus those data sets can be made available on a common data platform (e.g. Pangea, NSIDC). Some of the data sets have been processed by the authors (e.g. the accumulation radar data if I understand correctly) and would not be available to the scientific community if not published along with the paper.
AC: We believe a common data platform where all these data is gather is not needed. Most datasets are obtainable through the corresponding authors, and some are already distributed online (e.g. Favier et al., 2013). To accommodate the reviewer request we have elaborated on this in the data availability section (Section 7), noting where and how these data could be obtained:
*" All data used in this study are available without conditions by contacting the corresponding authors.*
*– RACMO2.3p2 model data (this study). Contact: j.m.vanwessem@uu.nl, m.r.vandenbroeke@uu.nl.*
*– AWS SEB data (Van Wessem et al., 2014a). Contact. c.h.tijm-reijmer@uu.nl, m.r.vandenbroeke@uu.nl.*
*– In-situ SMB observations (Favier et al., 2013). Contact: publicly available.*
*– 10mSnow temperature observations (Van den Broeke, 2008; VanWessem et al., 2014a). Contact: j.m.vanwessem@uu.nl, m.r.vandenbroeke@uu.nl.*
*– Accumulation radar-derived annual accumulation fluxes (Medley et al., 2015). Contact: brooke.c.medley@nasa.gov.*
*– GRACE mass anomalies (this study). b.wouters@uu.nl.*
*– Drifting snow transport fluxes (Amory et al., 2017). Contact: charles.amory@uliege.be.*
*– QuikSCAT meltfluxes (Trusel et al., 2013). Contact: trusel@rowan.edu.*
*– Cloudsat-CALIPSO (Van Tricht et al., 2016a). Contact: S.Lhermitte@tudelft.nl.*
*– Kohnen radiosonde data (this study). Contact: Gerit.Birnbaum@awi.de.*
*– Antarctic Peninsula ice discharge (Wuite et al., 2015; Hogg et al., 2017). Contact: Jan.Wuite@enveo.at."*

RC: Detailed Comments: In general, form of presentation and language are very good and the paper reads very well. A few smaller things are noted below. Figure 5 and corresponding discussion: Looks like the improvement in bias and RMSE is only because a few rather warm temperatures have been simulated better but this in a temperature range, in which the data show a lot of scatter. However, the new simulations clearly show a larger bias at low temperatures and this may be worth mentioning?
AC: We argue at several places in the manuscript (Sections 3.1, 3.2, 3.4.2, 3.4.3) that the larger bias at lower temperatures (in temperature, but also in precipitation) is due to the increase of clouds over the East Antarctic plateau, which is likely slightly too large. Overall, AIS temperature has gone up almost uniformly, which is reflected in the slight decrease of the bias. The reason that coastal temperature have increased by a larger amount than the

interior temperatures is likely due to a combination of feedbacks and processes that are too detailed to discuss in the scope of this manuscript.

RC: Can you give a reason why you do this elevation binning in Figure 8 instead of showing a map?
AC: As is described in Van de Berg et al., 2006 and Van Wessem et al., 2014, this binning has been done to distinguish between regions with a dense set of observations, and regions that have a limited amount of observations. The error bars reflect this effect.

RC: Section 3.5.4: Appears that you already start the comparison between coarser and higher-resolution model runs here; somewhat confusing since this is revisited in Section 4?
AC: Indeed, we have moved the small part discussing the RACMO2.3/AP snowmelt values to section 4.1.

References: Amory, C., A. Trouvilliez, H. Gallee, V. Favier, F. Naaim-Bouvet, C. Genthon, C. Agosta, L. Piard, and H. Bellot (2015), Comparison between observed and simulated aeolian snow mass fluxes in Adelie Land, East Antarctica, Cryosphere, 9(4), 1373-1383, doi:10.5194/tc-9-1373-2015. Dai, X., and N. Huang (2014), Numerical simulation of drifting snow sublimation in the saltation layer., Scientific reports, 4, 6611, doi:10.1038/srep06611. Vionnet, V., E. Martin, V. Masson, G. Guyomarc'H, F. Naaim-Bouvet, A. Prokop, Y. Durand, and C. Lac (2014), Simulation of wind-induced snow transport and sublimation C3 in alpine terrain using a fully coupled snowpack/atmosphere model, Cryosphere, 8(2), 395–415, doi:10.5194/tc-8-395-2014. Zwaaftink, C. D. G., R. Mott, and M. Lehning (2013), Seasonal simulation of drifting snow sublimation in Alpine terrain, Water Resources Research, 49(3), 1581-1590, doi:10.1002/wrcr.20137

[Figure]

*Figure 1: Monthly horizontal snow drift transport (a) and 10 m wind speed (b), RACMO2.3p2 (blue) and RACMO2.3p1 (red) for the year 2013.*

**Anonymous Referee #2**

RC: This is a high-quality comprehensive evaluation of the revised version of RACMO2.3 model over Antarctica in relation to its prior formulation for a wide variety of variables. The use of AWS and upper air data is limited however. Overall the improvements are modest but most notable in the surface mass balance components. There are a small number of issues.
AC: We thank the referee for these positive comments. We will address the small number of issues below:

RC: 1. Page 2, line 2: Many, many authors have discussed expected Antarctic warming is accompanied by increased snow fall estimates, so please add 1-2 more references. One recent example is Palerme et al. (2017) on CMIP5 models in Climate Dynamics.
AC: We added this reference.

RC: 2. Page 2, Line 26: Drop "so-called".
AC: Corrected

RC: 3. Page 2, Line 47: The atmosphere does not interact with topography rather its motion is impacted by topography.
AC: Corrected.

RC: 4. Page 2, Line 63: airborne elevation data - do you mean from Icebridge? Otherwise contemporary amounts must be very small.
AC: We cite the GLAS/ICESAT DEM here, which is, among other datasets, included in the source DEM for RACMO.

RC: 5. Section 3.4.2: Remind the reader that the 10-m firn temperature closely matches the annual mean 2-m air temperature. By the way, what is Ts?
AC: Corrected in the data/methods section to: "*For modelled surface temperature (T_s) evaluation, we use the 64 10 m snow temperature observations (Fig. 1) used in Van Wessem et al., 2014, which at this depth are representative of annual average surface temperature).*" The abbreviation is introduced in this sentence as well.

RC: 6. Figure 4: Are these values monthly means?
AC: Yes. Corrected.

RC: 7. Figure 7: Present the biases as well as the RMSEs. Discuss in the text.
AC: We chose to only present a figure with the RMSE to limit the amount of total figures. We believe the qualitative improvement is obvious from the figure

---

## Author Response (AR2)

Interactive comment on "Modelling the climate and surface mass balance of polar ice sheets using RACMO2, part 2: Antarctica (1979–2016)"

**M. Lehning (Referee)**

RC: Dear authors

Most review comments have been answered and I thank you for the revision effort. The model explanations are still too general in my opinion: For example, why not presenting the equation in which the cloud parameter l_crit is used along with an explanation of the context?

AC: We thank the referee for his final comments. We have added the cloud parameters equation for the onset of precipitation generation to the manuscript and a short explanation of its terms.

*"Therefore, the critical cloud water and cloud ice content ($l_{crit}$) threshold, that governs the onset of effective precipitation formation for mixed-phase and ice clouds, is increased in the following equation, adapted from ECMWF-IFS (2008):*

$$G_{precip} = Ac_0 l_{cld}\big[1 - \exp\big\{ - (\frac{l_{up}}{l_{crit}/\mathrm{BF}_e})^2\big\}\big].$$

*Here, A is a scaling value which is the cloud fraction for stratiform clouds and the updraught strength for convective clouds, respectively; $c_0$ is the coefficient for autoconversion of cloud ice/water into snow/rain; $l_{cld}$ the total cloud ice and water content and $BF_e$ an enhancement factor for stratiform mixed phase clouds. The value of $l_{crit}$ is increased by a factor 2 for convective clouds and stratiform water/mixed phase clouds, and by a factor 5 for stratiform ice clouds. "*

RC: I still encourage the authors to publish the observational data set collection en block with the paper. Just think back how much work it was to collect those. It is not helping open science if somebody has to do this again. Persons and emails will change, a published data set is a more stable milestone.

AC: We have uploaded available datasets on a public server if they were not freely available elsewhere. Some data sources request to be contacted by email, so for those datasets we have kept the previous description in the data-accessibility section. The section now reads as follows:

*"The following data are available through the IMAU website: http://www.projects.science.uu.nl/iceclimate/:*

- *RACMO2.3p2 model data (this study). Contact: j.m.vanwessem@uu.nl, m.r.vandenbroeke@uu.nl. – AWS SEB data (Van Wessem et al., 2014a). Contact: c.h.tijm-reijmer@uu.nl, m.r.vandenbroeke@uu.nl.*
- *10 m Snow temperature observations (Van den Broeke,2008;Van Wessem et al.,2014a). Contact:j.m.vanwessem@uu.nl, m.r.vandenbroeke@uu.nl.*

- *GRACE mass anomalies (this study). Contact: b.wouters@uu.nl.*
- *Drifting snow transport fluxes (Amory et al., 2017). Contact: charles.amory@uliege.be*
- *Cloudsat-CALIPSO (Van Tricht et al., 2016a). Contact: S.Lhermitte@tudelft.nl.*
- *Antarctic Peninsula ice discharge (Wuite et al., 2015; Hogg et al., 2017). Contact: Jan.Wuite@enveo.at.*

*All other data used in this study are available without conditions by contacting the corresponding authors.*

- *In- situ SMB observations (Favier et al.,2013). Contact:publicly available:http://www-lgge.ujf-grenoble.fr/ServiceObs/SiteWebAntarc/database.php.*

- *Neumayer meltfluxes (this study). Contact: s.l.jakobs@uu.nl, m.r.vandenbroeke@uu.nl.*

- *QuikSCATmeltfluxes(Truseletal.,2013). Contact:trusel@rowan.edu. Available throughQuantarcticav3http://quantarctica. npolar.no/*

- *Accumulation radar-derived annual accumulation fluxes (Medley et al., 2015). Contact: brooke.c.medley@nasa.gov.*

- *Kohnen radiosonde data (this study). Contact: Gerit.Birnbaum@awi.de. "*

The link on the IMAU website is still under construction, but the correct link will be provided when the peer review is finished and the manuscript will be finalized.